# Conceptual Change of 'Teaching' among Experienced Teachers after Studying Attentive Teaching

## Yaron Schur [1] and Ainat Guberman [2,*]

1 Attentive Teaching Institute, David Yellin College of Education, Maagal Beit Hamidrash 7, Jerusalem 9634207, Israel; schurfa@netvision.net.il
2 Teaching and Learning Department, David Yellin College of Education, Maagal Beit Hamidrash 7, Jerusalem 9634207, Israel
* Correspondence: ainat@macam.ac.il

**Abstract:** One of the obstacles preventing change of teaching methods in schools is teachers' traditional conceptualizations of 'teaching' as transmissive and teacher-centered. The aim of this study was to track changes in experienced teachers' concept of 'teaching', following their exposure to attentive teaching. This is a dialogical method in which the learners represent their concepts in drawings and written explanations, and discuss them with their teacher and peers. Method: This was a multiple-case study. The participants were three teachers who attended an attentive teaching professional development course. They drew 'teaching' in the first, sixth, and the last, fifteenth, session, and provided explanations of their drawings. Findings: At the start of the course, they described teaching as a unidirectional process of transmitting knowledge. In the middle, they became more aware of the students as individuals who should be listened to. By the end of the course, teaching was portrayed as multi-directional (and enjoyable), so that all the participants, including the teacher, teach and learn from each other. Conclusions: This study shows that by studying, experiencing, and implementing attentive teaching, it is possible to change experienced teachers' traditional beliefs without directly challenging them, and that drawings can track the changes' trajectory.

**Keywords:** attentive teaching; conceptual change; teachers; teachers' beliefs; teaching and learning; visual representations





## 1. Introduction

In Israel and worldwide, educational systems have reached the conclusion that in order to prepare students for the future, the traditional teaching approach in which learners have to acquire a given body of knowledge, is no longer sufficient. Since knowledge is changing and developing, teachers are required to foster independent learners who possess diverse learning strategies, use higher-order thinking patterns and are capable of collaborative learning with their peers. Therefore, teachers have to familiarize themselves with advanced teaching methods and implement them as part of their professional identity [1]. The problem is that the teachers themselves had often studied in traditional teacher education programs and conceptualized optimal teaching and learning according to how they were taught. These conceptualizations and beliefs concerning teaching tend to be impervious to change, even after teachers participate in teacher education programs that promote advanced teaching methods [2–4]. The aim of this study was to track changes in the concept of 'teaching' among experienced and leading teachers, after they were exposed to attentive teaching, a teaching method derived from social constructivism. First, we provided a brief explanation of social constructivism and attentive teaching, followed by a description of teachers' conceptualizations of 'teaching', and the challenges involved in changing them.

## 2. Social Constructivism

Social constructivism is today one of the leading approaches in teaching and learning. It developed from the teachings of Piaget [5], Dewey [6], Bruner [7], Vygotsky [8], Gardner [9], Feuerstein et al. [10], and others. According to this approach, learning is a process of construction in which learners' conceptualizations change when they link new knowledge to their previous knowledge. The learners acquired their previous knowledge in formal learning frameworks, or in informal settings through their daily experiences. The process of conceptual construction is an active one that requires cognitive effort because their previous conceptualizations may not be appropriate for their new knowledge and experiences. In such cases, the learners will have to reshape their previous conceptualizations, and not merely expand them by adding new facts. The term social constructivism emphasizes that the learning and conceptualization processes do not take place only in the learners' individual cognitive system, but they are also rooted in the social–cultural context. The language, conceptualizations, tangible world, and experiences that children and adults are exposed to, and the mediating dialog that organizes the world, were shaped throughout history by the culture they are living in [8].

Social constructivism describes learning processes and their influencing factors in general terms, but is not in itself a specific teaching method. Nevertheless, it is linked to student-focused methods, since they are the ones who have to construct the new conceptualizations. The teachers' role is to assist the learners in acquiring new knowledge and forming conceptualizations that fit accepted knowledge within their culture. The learned concepts will also be used by the students in extra-school activities and will form the basis for additional learning and future changes. Therefore, the teachers' role also includes helping the students acquire learning and cognitive skills that will help them acquire new knowledge and change existing conceptualizations in the future [1]. Several teaching methods are associated with social constructivism, including exposure to experiences that undermine previous "intuitive" perceptions, that do not fit the accepted knowledge in the culture; mediation; presentation of challenging assignments that require higher-order thinking skills; and actively involving students in discussions in which they raise ideas and critically examine them [11,12]. Teaching methods associated with social constructivism were found to lead to meaningful learning and a deeper understanding of the subject matter [1,13].

## 3. Basic Concepts of Attentive Teaching

Attentive teaching [14,15] is a teaching method derived from social constructivism. Concepts are learnt through dialog between learners and the teacher, as well as among the learners themselves, as opposed to a unidirectional presentation of the learnt matter by the teacher. The basic unit of attentive teaching is the mediated interaction. Vygotsky [8] described the mediation process as a dialog between an experienced adult and a learner that enables the learner to reach new understanding that is closer to the accepted understanding in society. Feuerstein et al. [10] emphasized that teachers' positive and supportive attitude is important in order to enable learners achieve cognitive change.

Interactive classroom mediation. In attentive teaching, the process of mediating between how the concept is understood by each of the learners, and how it is commonly understood in the culture of the learners occurs during the lesson in which the teacher addresses each and every learner. The mediation begins with the learners representing what they understand of the learned subject though a multi-modal text, i.e., a combination of a visual representation (illustration) and written explanation. They then present their outputs to their classmates and teacher who listen and comment as a discussion develops. The teacher's role in the dialog is to enable all the learners to individually express their knowledge and strong points, the world they come from, their unique point of view about the learned subject, as well as their doubts and questions. In this unfolding dialog, all the learners are required to be active, deepen their thinking process, and respond to their peers in a critical, yet respectful manner. In this way, the learners become partners in the teaching.

Each learner listens intently to their peers and makes serious comments, making them more aware of the importance of their personal expression and their contribution to the learning [16]. The teachers are active participants in the dialog. They adapt their teaching methods to the individual needs of each learner, and of the entire class, and keep track of the personal and unique developmental processes of the learners. With the teacher's help, the entire class forms a shared understanding of the learned subject. This shared understanding draws all of them closer to the learned concepts, as they are accepted and understood in the culture. Vygotsky [8] terms these as "scientific concepts". The unique individual concepts of the learners come together to form the basic principles common to the entire class. Thanks to the learners, the learned subjects become connected to worlds and fields that differ from those of the teacher, and thus the teacher's understanding is also expanded and refreshed [15].

The goal of attentive teaching is to enable students to experience conceptual change through a series of mediated interactions. We present two kinds of such series: dynamic learning and the thinking journey.

Dynamic learning—is a change observed in the learners' conceptualizations over time [14,15]. Attentive teaching attributes great importance to exposing the learners' intuitive conceptualizations before they begin their study of the learned matter, and to tracking how the conceptualizations change over time. The process of conceptual change is a very demanding one since learners construct concepts over time and they have become solidly entrenched [17]. When learners are exposed to scientific concepts (including concepts from the world of Humanities and Social Sciences) in formal learning frameworks, they tend to interpret them according to the intuitive concepts they formed during informal learning [8]. This process is sometimes called "mistaken conceptions" or "alternative conceptions" [18]. The learning is usually gradual, so the intuitive insights are not immediately replaced by scientific insights, but rather, unstable, intermediary concepts are formed until a new equilibrium of understanding is reached [19,20]. In attentive teaching, the visual representations of the learners' conceptualizations enable the teachers to observe the conceptualization processes in real-time during the lesson as a link is formed between the learners' world and their conceptual perception. Over the course of several lessons, the conceptual change in the classroom and in specific learners can be observed. Attentive teaching is implemented in diverse content worlds, beyond the world of scientific concepts. Each individual's unique concept is of great interest, and not only those parts that are connected to the accepted understanding of the concept in the culture, but also the parts that are connected to the worlds of the students [14].

Thinking Journey—is a process of conceptualization based on understanding the concept or learned subject in different contexts and from unexpected points of view [15,21–24]. According to Marton et al. [25], varied experiences over time are required to understand challenging subjects. Each of these experiences creates a connection between the learners and a different aspect and point of view about the learned subject. These observations create a "space of learning". The accumulation of the different points of view invites learners to make comparisons, to emphasize the similarities and differences between the different sightings of the learned subject and enables them to create a broad and coherent perception of the learned subject.

Carey [17] claimed that a space of learning is not enough for creating conceptual change. She stated this change can only occur through observing the familiar environment from a new and unfamiliar point of view. By changing their point of view, learners become aware of the limitations of the existing concept and of the need to expand it so that it becomes appropriate for additional contexts. The thinking journey method provides learners with opportunities to observe the world from different points of view to those they are accustomed to, thus implementing the insights of both Carey [17] and Marton et al. [25]. The teachers' task is to choose those points of view for the learners that will help them reach conceptual change by comparing the regular and familiar contexts with those chosen by the teachers.

Uncertainty processes in learning—are the emotional processes the learners undergo as they experience change. These emotions range from curiosity to feeling uncomfortable. When experienced in the right measure, the motivation to learn increases. In attentive teaching, uncertainty processes derive from the need to deal with leaving behind what is familiar, and with the need to change oneself, i.e., to do things they have not done previously and to connect these experiences with a new field of knowledge and to unfamiliar contexts [15]. The emotional component is an inseparable part of the entire learning process because when there is no emotional investment, the learning does not yield structural or intellectual change [26].

## 4. Using Multi-Modal Texts to Study Learners' Conceptualizations

Tracking learners' conceptualizations and how they change is a challenge. When asked to represent their conceptualizations orally or in writing, learners tend to regurgitate texts they have previously read or heard without really understanding them [27]. One of the ways to discover learners' concepts is through their drawings. This method is common in science teaching [28–30].

In attentive teaching, this method is expanded to be used in all subjects, at all ages. Drawings enable learners to express their unique point of view on the learned matter. The many advantages of presenting knowledge through drawings include: (1) Drawings are suitable tools for expressing conceptualizations. In a drawing, learners have to choose the elements they feel are the most important and to organize the overall relationships between these elements in an efficient way, while using the entire paper as an organizing tool and size the elements as an expression of their relative importance. (2) When preparing a drawing, the learners are required to be active and involved; to choose key elements, to decide how they should be presented, to clarify vague points, and to give reasons for the decisions they made. Verbal information that is passively repeated by rote is insufficient for producing a drawing. (3) A drawing is a communication tool. The fact that the drawing explicitly represents concepts makes it a comfortable springboard for critical examination and dialog about the clarity and coherency of the presented contents, and the extent to which they are compatible with the accepted perceptions. (4) A drawing is a learning tool. Since many pieces of information are organized in a clear and concise manner, it encourages questions to be asked, conclusions to be drawn and conceptual change. (5) In a drawing, the learners can personally express what they have learned. They link the learned matter with their personal worlds. Even in the classroom setting, the learners express their unique point of view in their drawings [15,31,32].

However, a drawing alone cannot shoulder the burden of transmitting meaning because the observers may not fully understand how the various elements in it should be interpreted and the specific relationships that exist between the different elements. Therefore, it is often necessary to combine visual images with written or verbal language. The resultant text is multi-modal, since different modes (in this case, images and written language) jointly convey a single communicative message [33,34]. When teachers examine the drawings and listen to the verbal explanations, they are able to understand the individual and distinct way that the learners understand the learned subject. The teachers can then adapt their teaching to meet the learners' needs precisely [15]. The very fact of asking the learners to draw a picture ensures that the teachers have to think about the key concepts that the learners should be learning. Repeating the drawing and explanation task throughout the teaching of the learned subject enables teachers to track the learners' conceptual changes. In attentive teaching, teachers methodically and widely track learners' conceptual change (dynamic learning).

## 5. Teachers' Conceptualizations of 'Teaching'

Teachers' perceptions about teaching and learning are important since they influence their targets, practices, and self-evaluation (but in a complex, rather than direct and simple manner). These perceptions also influence teachers' engagement in professional learn-

ing opportunities [35]. The perceptions are generally divided into two basic categories: traditional versus constructivist. According to traditional perceptions, the teaching process is a unidirectional process of transmitting knowledge from teachers to students, in which students play a minimal role, confined mainly to obedience, listening, and practice exercises. The main responsibility for teaching and learning is placed on the teachers' shoulders. In contrast, according to the constructivist approach, students are at the center of the learning process [35]. In reality, however, researchers often find that teachers have mixed beliefs; some of which are more closely aligned with the traditional, teacher-centered approach, whereas others are more student-centered. 'Mixtures' of different beliefs are prevalent among teachers with different amounts of experience, before, during, and after participation in intervention programs [36].

Traditional perceptions of teaching are difficult to change, even when intervention programs take place over long periods of time [2,3,37]. This could be explained by the fact that as students, teachers had only been exposed to traditional teaching methods and built their successful teaching careers within such systems. Social and cultural factors are also involved [2,37–39]; teachers' beliefs are less likely to change when they are long-held and connected to other ones that teachers espouse [4,35,36].

Girardet [4] conducted a literature review in order to find characteristics of intervention programs that enhance change in teachers' traditional conceptualizations. She described four factors: (1) reflecting upon and challenging teachers' prior beliefs, (2) studying alternative practices, (3) learning by doing, (4) collaborative learning with colleagues. Attentive teaching incorporates critical reflection and discussion among colleagues as an integral part of the method. The intervention program described below included experiencing attentive teaching as the program's method of teaching, and implementing it (learning by doing), thus incorporating all four factors. However, until now, the impact of attentive teaching on teachers' conceptualizations has never been empirically tested in areas other than the natural sciences.

Identifying teachers' conceptualizations of teaching and conceptual change processes is a persistent challenge since teachers are often unaware of them and cannot describe them verbally. One of the methods that attempts to overcome this challenge is asking teachers to draw what good teachers do and to add verbal explanations. In addition to the above-mentioned advantages of drawings in presenting concepts, they also bring to the surface the implicit and emotional aspects of teachers' beliefs, enabling them to express their opinions in their own unique ways. Verbal explanations help researchers understand the meanings teachers ascribe to their drawings and complement the meanings the drawings convey [40,41]. Drawings that convey transmissive conceptualizations of good teaching often focus on teachers in a traditional school environment. Students, if they appear at all, are relatively passive, or are all engaged in the same activity [40]. Teachers who adopt traditional attitudes convey mixed feelings in the texts they produce [41]. In contrast, drawings that convey student-centered conceptualizations depict active students in a central position. The students may collaborate with each other or engage in different activities. The teacher, if portrayed, is in the same area as the students and engaged in the same activities [40]. The emotional tone of the drawings tends to be positive [41].

Most of the studies that track changes in teachers' beliefs evaluate them along the student- versus teacher-centered dichotomy, based on two measurement points, in different frameworks such as before and after teacher education studies, gaining teaching experience or participation in an intervention program. Therefore, a more nuanced study that tracks changes in teachers' beliefs over time is needed [35,36].

In order to address both these gaps, this study aimed to understand how experienced teachers' concept of teaching changed after studying, experiencing, and implementing attentive teaching, by examining their drawings of 'teaching' in the beginning, middle, and end points of the course.

### 6. The Context of the Study

The current study was conducted as part of a 60 academic hour (15 meetings) professional in-service training course held in 2022 for female teachers at an ultra-Orthodox girls' seminary. The participants had many years of experience in frontal teaching of Jewish studies to large classes (approximately 45 students per class) in traditional frameworks. The training course had three goals: to present them with the attentive teaching method in an experiential manner, to bring about a change in traditional perspectives on teaching and learning, and to enable the participants to apply the principles of attentive teaching in their classes. During each meeting, there was a mediated interaction. The participants drew their conceptualization of the subject discussed in that particular meeting, added a written explanation to their drawing, and presented their individual point of view to the other participants. Each participant was asked to give her opinion on her peers' outputs, and about the subject being discussed. The discussion was respectful and attentive so that a shared understanding of the subject was formed. The first and main section of the course was 'dynamic learning' about the concept 'teaching'. At the beginning (1st meeting), middle (6th meeting), and end of the course (15th meeting), the participants drew a picture and added a written text to explain how they understood this concept. This enabled us to track changes in this concept. The first section of meetings in the in-service training course (meetings 2–5) was a 'thinking journey' that observed 'dialog' and 'listening' from different points of view. The second meeting dealt with 'listening'. Working in pairs, each participant was asked to relate an experience, and then to listen to her partner's experience. They were then asked to draw two pictures with a written explanation. The first portrayed them as a listener and the second as a narrator. The third meeting focused on classroom dialog, inspired by a well-known poem "Two Elements" by Israeli poet Zelda that describes a conversation between a flame and a cypress tree. The poem can be viewed as a metaphor for communication difficulties in the classroom that is open to different interpretations concerning the teacher's role in bridging over them. The fourth lesson was about dialog with a challenging student, presenting various examples about how to relate to students who do not keep to the 'straight and narrow' path, how to see their strong points, and to find opportunities to let them show these points. In the fifth meeting, they discussed their experience as learners: inspirational teachers who successfully formed meaningful relationships with them. They described impressive teachers who were able to reach them and leave their mark. Whereas in the discussion about challenging students, the dialog focused on empowering such students, when the participants were discussing their own personal experience, the focus was more on the image of the teacher. Which teacher actually left her mark? How did she function? And, under what circumstances did she manage to reach the students? A discussion evolved about the personal example that teachers set in their behavior, and in how they related to the learners. The various points of view about teachers' behavior and about different aspects on teacher/student dialog, as well as ways of forming an attentive dialog all formed a thinking journey about listening processes in the classroom.

The ensuing meetings used dynamic learning to track change processes in understanding a Talmudic text. The participants learnt how to prepare lesson plans using the principles of attentive teaching, and applied them in their own classrooms as well as in the in-service training course. Some of the lessons were taught by the lecturer, and others by the participants. The participants taught lessons about Jewish festivals, history, Bible, Talmud, and science and shared what they learnt about the subject with their peers and the lecturer. For example, one participant chose to teach "The Binding of Isaac" which was part of the 11th grade syllabus. In the spirit of attentive teaching, she asked the learners to imagine themselves being tested as if they were Abraham who was being tested by God. One of her students, who was vision impaired (borderline blindness), wrote a poem in which she described her feelings. She wrote an imaginary discussion with God, full of respect and faith, about her condition. The mediated interaction about a "test" helped

the teacher and students to get to know each other better and to link their world with the concepts being studied in an experiential manner, and not just in an abstract form.

## 7. Method

This was a multiple-case study that strives to understand a small number of cases within their real-life contexts [42]. The participants were three of eight female Jewish-studies teachers who attended an attentive teaching in-service training course. Their ages ranged from 35 to 55, and their years of teaching experience ranged from 10 to 35 years. We asked the course participants to draw 'teaching' and add written explanations of their drawings during the first, sixth, and final (15th) meetings, so that we could track the changes in their conceptualizations of 'teaching'. After we received permission from the institutional research ethics committee, and after the in-service training had ended, we contacted the course participants and asked their permission to study the materials they produced during the course. They all gave permission, but here we present data from only three of them. The three participants were randomly selected. We limited the number to three believing it was small enough to allow us to provide detailed descriptions of the data analysis, and large enough to show the similarities, that were in fact shared by all eight participants. The participants' names were anonymized to protect their privacy.

The materials we analyzed were drawings and written texts describing the participants' conceptualizations of teaching at three time points: the start, middle, and end of the course. During the analysis, we referred to each drawing and accompanying written text as a multi-modal text [34], and examined the following aspects: what metaphors did the participants use, and how did they position the teachers' and students' respective roles and relationships? [43]. Was the teacher presented and what was the size of the teacher's image in relation to other images and the whole drawing? Large size images convey importance [44] and are associated with teacher-centered views [40]. What were the teachers doing? Lecturing, explaining, and exemplifying are characteristic of transmissive conceptualizations, whereas in student-centered drawings the teacher (if he or she are presented), is engaging in the same activity as the students [40]. Teachers' actions were identified based on the images as well as on the explanatory texts. Did the teachers occupy the same space as their students, or were they separated from each other? Separation is an additional element that is associated with traditional views [40]. Were the teachers and the students connected by lines or arrows? The arrows' direction symbolizes the flow of influence and power [44]. As for the students, we asked whether they were represented, and what they were doing. Teacher-centered views often result in the omission of students or them being portrayed as passive, listening and looking, or engaging in the same activities [40]. How was the learning environment portrayed? Rows of desks, blackboards, and teachers' podium are associated with transmissive teaching [40]. Finally, we examined the facial expressions of the depicted characters and looked for emotions conveying expressions within the explanatory texts [41]. The study's two authors analyzed the drawings separately and then compared their analyses and reached agreement, in order to increase the reliability of the study.

## 8. Findings

In this section, we analyzed each drawing and written explanation, and uncovered the personal journey each participant of the in-service training course underwent. We then discussed the themes common to all the participants.

### 8.1. Miriam

Miriam taught Jewish studies in upper grades. She was very interested in the role of women in Jewish society and in educating her students to fulfill this role. She demonstrated the traditional perception of teaching, as seen in Figure 1 when she drew her conceptualization of teaching at the start of the process. The right side of her drawing shows the teacher's eye, heart, and speech, and the left side shows the student's eyes, heart, and understanding

(in the shape of a mental flash of understanding). The arrows point in the direction from teacher to student, representing a unidirectional flow from teacher to student [44]. The teacher transmits knowledge, shapes (a term that appears twice in her written text) gives, observes, perceives, talks, is responsible and ultimately successfully reaches her students' hearts. For Miriam, transmitting knowledge is the teacher's responsibility. As opposed to the teacher, the student in the written text demonstrates opposition, bitterness and anger. To mitigate the impression in the text of a "conflict", the drawing shows a smiling learner, and a connection between the hearts and the flash of understanding, all conveying positive emotions [41]. There was no representation of the class as a group, and no evidence of changes that the teacher experiences as a result of her teaching.

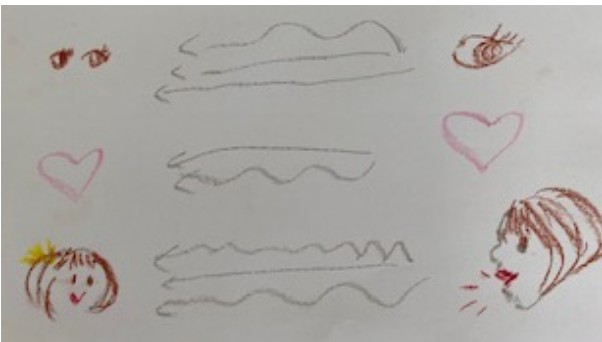

Text: In my view, teaching is a wondrous interaction between teacher and student … with the ability to impart knowledge, and especially to shape. To shape a path, trust, values, the soul… I believe that true teaching occurs along the path between what the teacher gives and what the student receives… it is the teacher who is responsible for creating this path. For me, the process is: observation and perception which lead to speech that creates a flash [of understanding] … for me this is an experience that should be observed, even during the lesson. …how I perceive the student … that I see that she sometimes is opposed, is bitter or angry. Do I manage to find the way to her heart? At the end of the day, I am imparting knowledge. It is important for me to transmit as much as possible …

**Figure 1.** Miriam at the start of the course.

At the sixth meeting, Miriam produced a second drawing in which she expresses the transition from the teacher's point of view to that of the learner. The student's figure is large and positioned at the center of the drawing, whereas the dotted lines that fall on the student's notebook represent the information the teacher provides [40]. Miriam is now concerned about the gap between what she is trying to transmit to what the student writes in her notebook. As opposed to the first drawing in which Miriam identifies with the need to "transmit" as much knowledge as possible to her students, in the second drawing Miriam expresses her discomfort with the system that forces her to reach goals in a given time frame. She exhibits relative change in that she now wishes to see each student as a unique individual, rather than having a goal of "shaping" them, in which no distinction is made between each individual. She also notes the contradiction between her wish to form a personal connection with each of her students and the judgmental approach that characterizes traditional teaching methods, and even criticizes the system she works in. The concept of teaching has changed for her, from a process in which the teacher is responsible for shaping an object to showing an interest in the student as an individual and giving up on "covering material" and being judgmental.

Miriam produced the third drawing after she had completed the 15-meeting course. She represents herself and her students as connected vessels who create understanding. The drawing does not only show the individual student, but also the entire class. The interactions are multi-directional in the connected vessels. The liquid in each of the test tubes (that represent the partners in the learning process) influence and are influenced by the other test tubes. All the students learn, as does the teacher who is represented by a slightly taller vessel than the ones representing the students. Engagement in the same activity as the students is typical of the student-centered approach, as is the depiction of the learning environment devoid of objects that are associated with traditional schools [40]. In the accompanying explanatory text, learning is a never-ending process. Instead of 'material' and 'values' that the teacher has to transmit within a given time, there is continuous learning. The teacher needs the students, and she herself is refilled because of them. She 'fills' and 'is filled', and without them she 'dries up'. The explanatory text expresses a calm, using positive words such as 'benefit', 'plentifulness', and 'developing' instead of mentioning opposition that forms obstacles. In both Figures 2 and 3 the image and written text are mutually reinforcing and complementary.

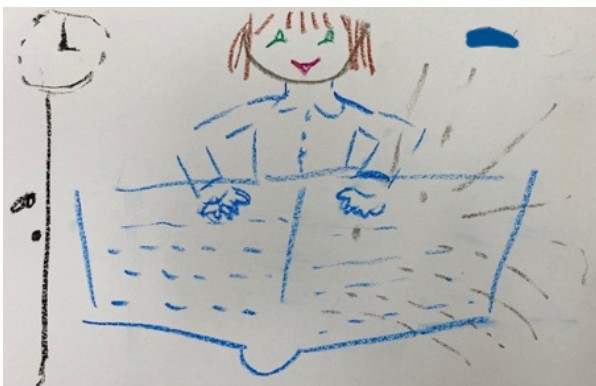

Text: I feel I am going through a process. I drew one student, not a whole class. To see each student for what she is. I hope that my words go beyond what she writes in her folder. The gap between a homeroom teacher and a subject teacher. The door and clock represent the requirements of the system. I have difficulty with the critical comments and the judgmental attitude.

**Figure 2.** Miriam half-way through the course.

Miriam is aware of the process she underwent and sums it up:

I myself experienced a process of patience and tolerance. I learned how to listen. To listen from a very real place. The person opposite me can teach me something new. I can learn from my students. They can teach me new things. They have what to say to me. I went through a personal process of listening and empowerment. My drawings, my thoughts and my insights are significant. A dynamic has been set in place, a surprising and enriching dialog is developing thanks to each person's involvement. And each person is part of this human process.

At the end of the in-service training course, Miriam's concept of teaching is now of a shared process of shaping understanding. It is a multi-directional process in which the students' role is no less than that of the teacher. They are all enriched from the joint learning, in a pleasant class atmosphere. At the start of the process, Miriam was very concerned about the need to cover all the study material, in the middle she criticized this need, and in the final text she did not even mention it.

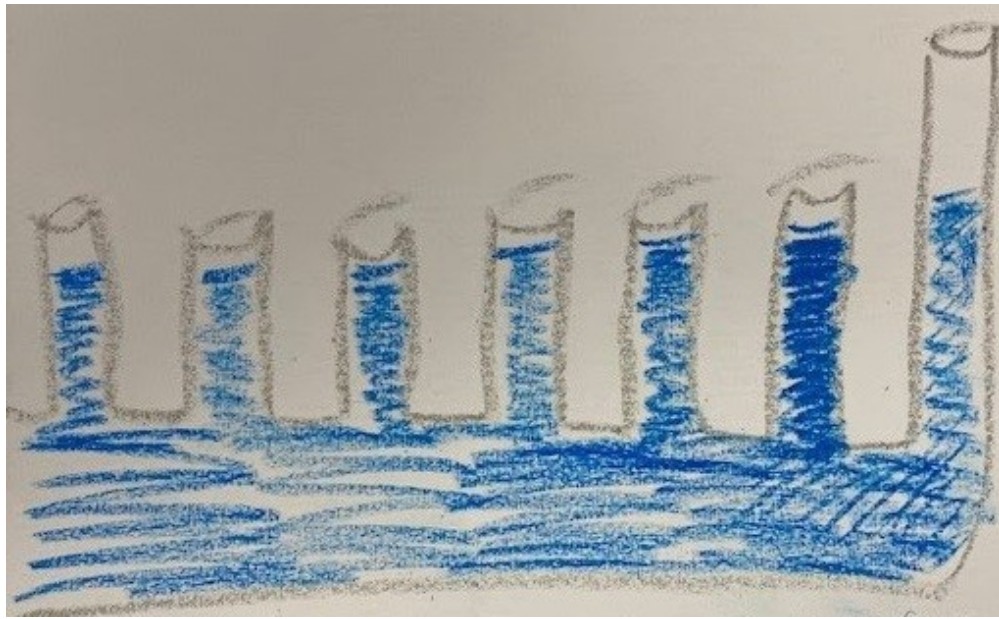

Text: I am going through a process—like connected vessels. All the lights that are connected to me, and all the vessels close to me benefit from this plentifulness and from this process. It is automatic. The more vessels there are, the better it is for all. In the drawing it is completely schematic and is also limited in the quantity of water. In practice—it is never-ending. When I develop, I have an influence on all those who are influenced by me. It is frightening to think that this is boundless. Frightening to think how quickly I dry up again and have to get wet again in order to be filled up again and to fill others up again.

**Figure 3.** Miriam at the end the course.

*8.2. Sara*

Sara was an elementary school principal, as well as a Jewish studies teacher. Figure 4 depicts how Sara perceived the concept of teacher at the start of the in-service training course. It shows a parachute propelling the students high up into infinity. The students and teacher are not represented in the drawing; however, the explanatory text makes it clear that the teacher is responsible for the teaching: "It is impossible to describe the extent to which teaching has the power to develop, give tools, enable, guide, have far-reaching consequences for learners". All these verbs describe the teacher, with the students having a passive role. The text also clarifies the meaning of the visual metaphor where the students are passive passengers carried by the wind. "Ways of connecting with each student and ensuring that no one is left behind are limitless—and one should never give up. You should always look for that extra teaching method, they one you haven't yet tried". According to the text, the teacher is looking for ways and means. She is charged with reaching each student and ensuring that no one is left behind. Like Miriam, Sara says that the responsibility lies with the teacher. Like Miriam, Sara also gently alludes to her moments of frustration where the teacher has to try and find new ways and not to give up [41].

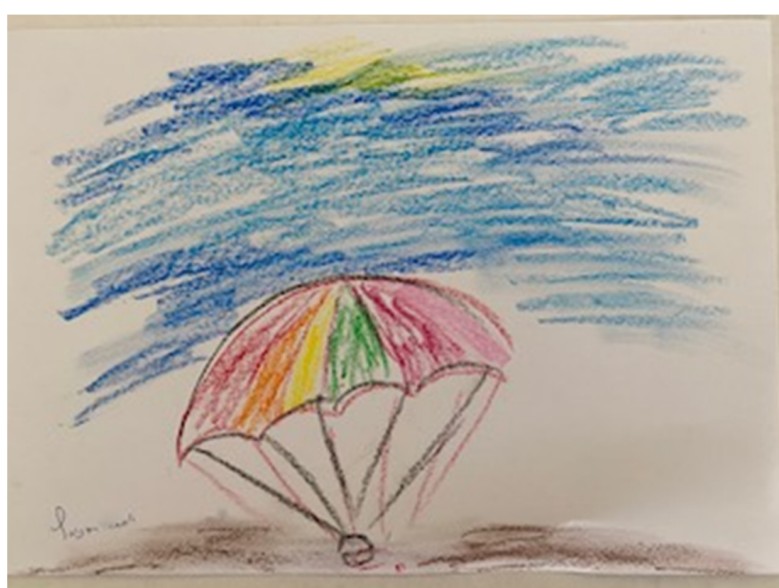

Text: … I thought that the phrase "the sky is the limit" could be an overall saying that encompasses everything that is included in teaching. Improving one's teaching practice, and the ability to have an influence through teaching is limitless. There is no end to what can be learned about teaching and from it. And as a student, one cannot begin to describe the extent to which teaching can give tools, can enable, direct, and have far-reaching consequences for students. Also, ways of connecting with each student and ensuring that no one is left behind are limitless—and one should never give up. You should always look for that extra teaching method, the one you haven't yet tried. In my drawing, the sky represents infinity, and the parachute flying up high is influenced by the wind, air etc.

**Figure 4.** Sara at the start of the course.

Figure 5 was drawn by Sara at the end of the process. The students are represented as colored pencils. Each one has her individual character and unique abilities to contribute to her peers and to the teacher. The written part of Sara's multi-modal text shows she believes that knowledge is a joint outcome shared by all. By listening to her students, the teacher is no less enriched than her students from the joint learning. Each student has her unique knowledge, and together, they form the wisdom of the crowd which we all benefit from. "And now we learnt about 'Attentive' [Teaching] and we learned: Pay attention, to something else, an additional, different, similar, close meaning. Someone is out there saying something that no one else has said!!! Amazing!" The exclamation marks and the word 'amazing' indicate excitement, which is very different from the sense of frustration that was gently mentioned at the start of the process. Like Miriam, at the end of the process Sara also viewed learning as a shared process in which all are mutually enriched. Both teachers experienced a similar change from the beginning to the end of the process. Sara did not participate in the sixth meeting, so we do not have any information on what she experienced in the middle of the process.

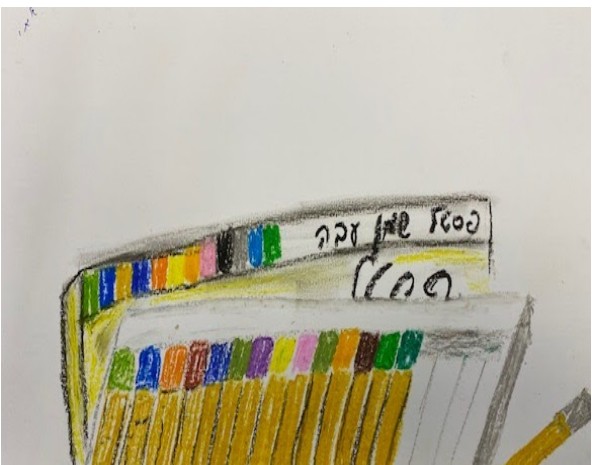

Text: Individuals each perceive and interpret every "objective" event or idea in different ways. No one idea is identical to a second one. They may be similar, close, or they may be the opposite or complementary. Yet, each idea is unique and emphasizes a message, insight or a specific statement. And it is an experience—to feel the difference in perception. Until now, we learnt that every concept has a lexical, scientific explanation which is considered a uniform, precise definition. And now we learnt about 'Attentive' [Teaching] and we learned: Pay attention, to something else, an additional, different, similar, close meaning. Someone is out there saying something that no one else has said. Amazing. What do I take away from this? The intensity of the wisdom of the crowd that I see even in a classroom of young students and the intensity of the knowledge that can emanate from the class! Listen to the students. They can teach all of us. "From all who taught me have I gained understanding, but from my students—more than from all of them"

**Figure 5.** Sara at the end the course.

*8.3. Rebecca*

Rebecca taught junior-high students with learning difficulties, as well as Jewish studies in mainstream classes. Figure 6 was drawn by Rebecca at the start of the in-service training course. The teacher is represented as a lifeguard, positioned at the center of high waves, and the students are invisible [40]. Like her two peers, Rebecca's first drawing presents the teacher as having sole responsibility for teaching, and the students (who are "drowning") are shown as passive people who need to be rescued. The arrows are directed to the drowning students in a top-down manner [44]. The emotional atmosphere is ambivalent. On the one hand, teaching leads to a sense of pressure and distress, yet on the other hand, it is therapeutic and soothing.

Midway through the course, Rebecca produced Figure 7, in which the teaching was once again at the center of the teaching process. However, the direction of the information flow had changed. The focus is the teacher who listens, and who sees her students. Knowledge transmission (as represented by the mouth) is in second place. The written text emphasizes that the teacher keeps silent and concentrates on listening. As opposed to Figure 6, in which the teacher is represented as the "lifeguard" whose actions are vital and justified, the explanatory text to Figure 7 shows that the teacher has to be completely frank and open-minded.

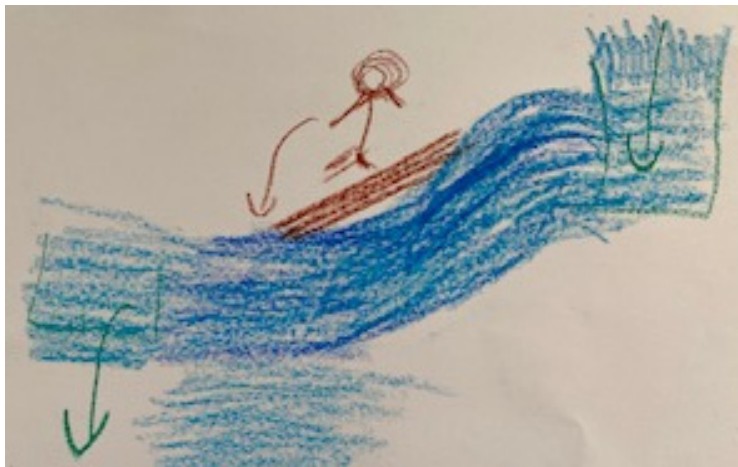

Text: The teacher is the person who calms others, a savior, the responsible adult. Teaching can cause distress and overflow, just like water but it can also be soothing. Learning is something therapeutic.

**Figure 6.** Rebecca at the start of the course.

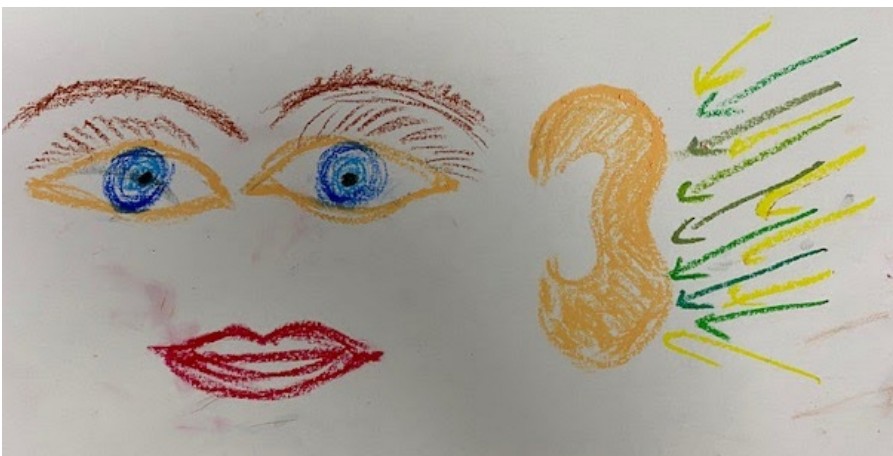

Text: Teaching—a large ear and big and beautiful eyes, and a mouth, that is important, but smaller and shut. The teacher hears much more beyond [what is spoken]. It is multi-tonal and moving. One can only see and listen if one is really sincere.

**Figure 7.** Rebecca half-way through the course.

Rebecca produced Figure 8 at the end of the in-service training that shows a train. Each wagon represents a different component of teaching: The first—question marks; the second—to see each and every student and to learn from them; the third—the ripple effect, to see the good and to pass it on; the fourth—to believe in the process. When the teacher is free and available to listen, the student is free and available to learn. In her explanation, Rebecca notes three important elements: teaching is a long-term process. Teaching and learning are conducted in a group—that enriches, expands, and contributes to the participants' understanding in various fields. The teacher learns from her students and undergoes a change process. The 'savior's' sense of confidence was replaced by question marks that drive the process, and with a sense of humility. The emotional atmosphere improved. Like Miriam and Sara, Rebecca was also excited by the process she underwent, and especially by the understanding that she could learn a lot from her students. Rebecca used the same Talmudic phrase, as did Sara: "From all who taught me have I gained understanding, but from my students—more than from all of them" (Ethics of the Fathers, 4:1) which demonstrates the classroom process and how the 'new' concept of teaching is connected to the world of the teachers who participated in the in-serving training course.

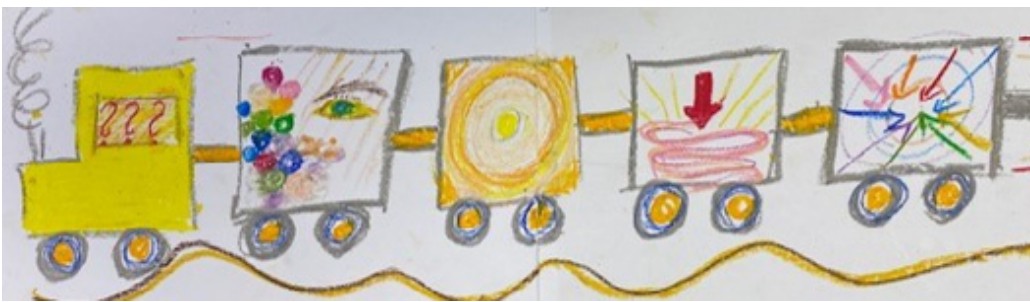

Text: I feel that the process is the title of the story. At the beginning, I did not really understand what was happening. There were a lot of question marks? Over time, I felt which direction it was taking and to what depths and places one can reach with "Attentive" [Teaching].

I am taking the following [ideas] to my teaching practice:

To see each and every student and to learn from the learners "from my students—more than from all of them", a sense of humility and trust in their inner being. To believe in the truth found in every person, and to show her the good, without criticizing. A good eye that impacts on the good going forward. When the teacher is available to listen, the student has an opportunity to learn.

I had an additional learning experience: a huge amount of enrichment from the diverse group of participants, this was such a special and enriching group. There were depths of thought, millions of emotions, understandings, frankness, sympathy, intelligence. To learn from the learners who were in the class with me.

**Figure 8.** Rebecca at the end of the course.

## 9. Discussion

The three sets of drawings show that each of the teachers had her own unique world of images about the concept of teaching. At the start of the process, a teacher talking to a student, a parachute that "propels the students up to the sky" and a lifeguard who prevents her students from drowning. In the middle of the process, two individuals are learning, one is studying from a text and the other uses all her senses. At the end of the process, connected vessels, a box of colors, and a train. Despite their diversity, the three teachers underwent a shared process. At the start of the course, teaching was conceptualized as a unidirectional process that is the sole responsibility of the teacher. The students react to the teacher and their reactions are sometimes frustrating. The teacher has to get the students to understand what she is trying to teach and overcome her sense of frustration with their lack of understanding or their opposition. By the middle of the process, the teachers had become more aware of the students as individuals who need to be listened to carefully. The teachers were now interested in the students themselves, and did not only view them as objects of their teaching practice. This is a challenging process involving criticism of the system they are part of. At the end of the process, the teachers' drawings referred to the entire class. They were interested in their students and were amazed by their knowledge. The teaching and learning process had become multi-directional so that all the participants, including the teacher, teach and learn from each other. The teachers became humbler, yet although their status may have seemingly been 'lowered', the atmosphere had become more positive and optimistic. The teachers were aware of the process they underwent and were very excited by it. They all pointed out the original ideas that their students provide. "What am I taking from it? The strength of the wisdom of the crowd, even when the students are young. What strength of knowledge can sprout in the classroom! Listen to the students; they all have something to teach us". Sara and Rebecca also quoted from Ethics of the Fathers: "From all who taught me have I gained understanding, but from my students—more than from all of them". The participants were used to teaching in large classes, and practice frontal teaching. They discovered that using the attentive teaching method, they could facilitate mediated constructivist learning processes, even in large

classes. They presented the lessons they taught and their students' outputs to their peers in the in-service training course about attentive teaching. The complementary texts to their drawings at the end of the process expressed their excitement about the method.

In four meetings, the participants had experiential discussions about various aspects of listening. They came to understand that when teaching, they only partially listened to their students, because they focused on 'transmitting' content knowledge. The meetings about listening were mediated interactions with active involvement of the teachers. They created multi-modal texts (consisting of drawings, written, and verbal language) to express their feelings about the concepts that were discussed in the meeting: listening; Zelda's poem "Two elements"; interactions with a challenging student; and an inspirational teacher. As a result of these mediated interactions, the teachers represented their students in a new way. They now became interested in each one's unique voice.

The second section of the in-service training course dealt with the construction of dynamic learning and thinking journey in units consisting of several learning meetings, as well as with analysis of conceptual change through drawings. As the teachers learnt about attentive teaching, they also experienced the various aspects of this method personally, since this was the teaching method used during this course. They were exposed to their peers' ideas and creativity and thus got to know their peers better and appreciate them more. Furthermore, they implemented the method in their own classrooms. At the end of the in-service training course, the participants' conceptualization of teaching had changed, and they recognized the students' contribution to their peers' learning, as well as to the teachers' learning. In previous studies, we examined attentive teaching's ability to generate change in natural science conceptualizations among schoolchildren and adult learners: weight and gravitation [23,24], the day–night concept [22], the planet Earth [14,21], and social-ecological systems [45]. In this study, we dealt with the conceptualization of 'teaching' among experienced teachers, Jewish studies teachers, who implemented attentive teaching in their classes—in subjects unrelated to the natural sciences. Therefore, the practical implication is that attentive teaching can be integrated into different disciplinary school subjects and teacher education contexts to bring about conceptual change in collaborative and enjoyable manner.

Our findings are similar to those presented in other studies that were successful in changing teachers' beliefs. Like the (student) teachers who participated in those studies, the experienced teachers in our in-service program started the process as teacher- and content-centered teachers engaging in transmissive practices. Similar to other successful intervention studies, we found that the combination of reflecting upon current practice, studying an alternative practice, learning by doing, and collaborative learning with colleagues could achieve significant changes in teachers' beliefs and actions [4]. At the end of the process, the participants adopted a social constructivist, student-centered approach, and expressed positive feelings about teaching [41,46,47].

There are also differences between those studies' findings and ours. Fives and her colleagues [35] (p. 250) pointed out that the tradition of dichotomizing teachers' beliefs into teacher- versus student- centered may be too broad to capture nuances and change over time. In our study, we measured teachers' beliefs three times during the professional development course and found out that the participants' change process occurred in three-phases. At first, the teachers held traditional views about teaching. Half-way through the course, they adopted student-centered views aligned with constructivism. The final phase that took place at the end of the course involved both teachers and students learning from each other, in line with social constructivism. In contrast with Yung-Chi [41], the positive emotions the teachers expressed were related to the discovery of the students as a source of knowledge and inspiration, and not merely from adopting student-centered practices. We hope that more studies that employ multiple assessments of teachers' beliefs longitudinally will enhance our theoretical understanding of the trajectories of change in teachers' beliefs.

In recent decades, interest in teachers' wellbeing has increased [48]. It has been established that teachers' wellbeing is related to their relationships with their

students [48,49]. It is possible that in our study, teachers' focusing on their students and particularly implementing Attentive Teaching during the second half of the course resulted in them gaining a higher level of appreciation for their students and learning from them. These effects may have improved the relationships between teachers and students even more, and as a result, teachers' well-being may have improved. It is also possible that learning from students makes a unique contribution to teachers' wellbeing. This possibility has theoretical, as well as practical significance and deserves further research.

Undermining learners' 'intuitive' concepts is often a part of constructivist ways of teaching [11,12]. This is accomplished by exposing students to experiences that raise cognitive conflicts, since they are incongruent with students' intuitions, but can be explained with the accepted cultural concepts and theories. Such practices do not lead to an immediate replacement of intuitive insights with culturally accepted ones, but rather to the formation of intermediate concepts that are positioned between the culturally accepted concepts and the initial ones the learners had prior to learning [19]. These are not stable, fixed concepts; the learners feel that their insights are not yet coherent and try to change them so that they are all in agreement [19,20]. Research on change in teachers' beliefs also found that teachers may have a mixture of beliefs, some of which are student-centered and others which are teacher-centered, and that these could be found before, during, and following interventions [4,35]. Instead of challenging learners' conceptualizations [11,12], attentive teaching provides them with mediated interactions that accentuate multiple aspects of the focal concept and enrich it (thinking Journey). As a result of participating in the course, the participants did not construct intermediate incoherent concepts [19,20], but rather coherent and richer ones that are more appropriate for actual teaching practice. Half-way through the process, the participants' conceptualizations were in line with constructivism and at the end of the process, with social constructivism. Both changes took place in a manner that respects and builds on the participants' prior conceptualizations. Enriching, rather than replacing, participants' prior ideas might explain how we achieved change, even though the teachers in our study belong to traditional communities, in which teacher- and content-centered forms of teaching are well entrenched and interconnected with other values. Such circumstances impeded change in other studies [4]. The teachers connected the new ways of teaching that the attentive teaching course introduced with aspects of their heritage, such as the Talmudic verse two of them cited. This finding suggests that a respectful approach towards learners' previous intuitions may be a better strategy for inducing change than creating cognitive conflicts, particularly while working in traditional contexts. Although this is not a novel understanding, it is both theoretically and practically significant given that creating cognitive conflicts is still a common constructivist practice [50,51].

Finally, we used drawings and their written explanations to track changes in the participants' conceptualizations of 'teaching'. This method was successfully applied in previous studies, either to document teachers' beliefs or to track changes from teacher-centered to student-centered beliefs [40,41,46]. By using the same technique, we were able to observe a three-phase change process. However, more studies are needed to discover additional indicators of different phases of change processes.

## 10. Conclusions

In this study, we aimed to find how experienced teachers' concept of teaching changed after studying, experiencing, and implementing attentive teaching, by examining their drawings of 'teaching' at the beginning, middle, and end points of the course. We found that at the start of the course, the participants were teacher- and content-centered using transmissive practices. Half-way through the course, they adopted student-centered views aligned with constructivism. By the end of the course, they believed that teaching was a multi-directional and enjoyable process that involved teachers and students leaning from each other. These beliefs are aligned with social constructivism. The change process took place even though the participants belonged to traditional communities, in which transmissive, teacher-centered practices are well entrenched and interconnected with other

beliefs, circumstances that are associated with resistance to change. Finally, the three phases of this change process were documented and identified through multi-modal texts consisting of drawings and written explanations.

Our findings suggest that attentive teaching is a practical method that can change experienced teachers' conceptualization of 'teaching' and contribute to their wellbeing. From a more general perspective, our findings suggest that extending and building on previous conceptualizations may be a better strategy for inducing change, rather than challenging these conceptualizations, particularly in traditional contexts that are resistant to change.

Our findings also point out three potential theoretical contributions that require additional research: (1) Identifying the three phases in changing teachers' beliefs about teaching, instead of the teacher- versus student-centered dichotomy. The third phase is associated with social constructivism. (2) The contribution that learning from students makes to teachers' wellbeing. (3) Using drawings and their explanations for nuanced and protracted documentation of teachers' beliefs and the changes they undergo over time. The study's limitations are its small scale and unique group of participants. Future studies will explore whether our findings can be replicated in additional cohorts, and follow these three directions of research.

**Author Contributions:** Conceptualization, Y.S.; Methodology, A.G.; Data curation, Y.S.; Writing—original draft, A.G. All authors have read and agreed to the published version of the manuscript.

**Funding:** This research received no external funding.

**Institutional Review Board Statement:** The study was conducted in accordance with the Declaration of Helsinki, and approved by the Ethics Committee of DAVID YELLIN COLLEGE OF EDUCATION on 2 January 2022.

**Informed Consent Statement:** Written informed consent has been obtained from the participants to publish this paper.

**Data Availability Statement:** The raw data supporting the conclusions of this article will be made available by the corresponding author on request.

**Conflicts of Interest:** The authors declare no conflict of interest.

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
