# Peer review of "Conceptual Change of ‘Teaching’ among Experienced Teachers after Studying Attentive Teaching"

_education, doi:10.3390/educsci14030231_

Round 1
Reviewer 1 Report
Comments and Suggestions for Authors
The authors are encouraged to continue to serve such quality work to their audience.

I am satisfied with the Language but maybe a proofreader can do a better job here.
Author Response
Comments 1: We are grateful for the positive evaluation of our work. There was only one comment that required revisions and it will be addressed below: “The Thinking Journey methodically applies both Carey and Marton’s insights” in lines 141 – 144 is unclear. |
Response 1: Thank you for pointing this out. We rephrased the sentence. Now it is: The Thinking Journey method provides learners with opportunities to observe the world from different points of view to the one they are accustomed to, thus implementing the insights of both Carey [17] and Marton et al. [25]. (Page 3, paragraph 5, lines 141-144). |
Reviewer 2 Report
Comments and Suggestions for Authors
I found this an extremely interesting and well written paper. It is well referenced, well structured, following an interesting research methodology and the findings highlight the changes in teachers' conceptions of teaching and the role of the teacher. Just a few minor comments: a) I don't think I would call the teachers (throse at least whose drawings you analysed) "veterans". I would call them experienced, b) a bit more information on the profile of the paprticipating teachers would be helpful, c) in the literature review section I think that reference to studies relating to teacher beliefs and how impervious they are to change would be helpful. There is extensive literature and research on teacher beliefs. Please go through the paper for minor typos. Apart from the above comments, I believe the paper is worthy of publication as is. Thank you for an interesting read!

Author Response
We are grateful to the reviewer for the positive evaluation of our work. The reviewer had four comments that will be addressed below. |
We are grateful for the positive evaluation of our work. There were four comments that required revisions and they will be addressed below. |
Comment 1: I don't think I would call the teachers (those at least whose drawings you analyzed) "veterans". I would call them experienced, |
Response 1: Thank you for pointing this out. We have replaced the term “veterans” with “experienced” in four locations where it previously appeared: The title, the abstract (lines 6 and 16), and page 5 paragraph 6 line 247. |
Comment 2: A bit more information on the profile of the participating teachers would be helpful. |
Response 2: The participants were female Jewish-studies teachers from the ultra-orthodox Jewish sector in Israel. They were used to frontal teaching of large classes (approximately 45 students per class). The participants attended an in-service training course in Attentive Teaching at the authors’ College of Education. Their ages ranged from 35 to 55, and their years of teaching experience ranged from 10 to 35 years. Miriam, the first participant, teaches in upper grades. She is very interested in the role of women in Jewish society and in educating her students to fulfill this role. Sara, the second participant, was also a school principal, whereas Rebecca, the third participant, was also a special education teacher. This information appears in the manuscript. We kindly ask what further information should be provided. |
Comment 3: In the literature review section, I think that reference to studies relating to teacher beliefs and how impervious they are to change would be helpful. There is extensive literature and research on teacher beliefs. |
Response 3: We introduced three new paragraphs to the section on teachers’ conceptualizations of teaching on page 5 and expanded the other two paragraphs of the section. |
Comment 4: Please go through the paper for minor typos. |
Response 4: Thank you for your time and effort. We corrected the typos and added clarifications where needed. A file with full details about the changes we made in accordance with your suggestions was uploaded to the journal’s site. |

Reviewer 3 Report
Comments and Suggestions for Authors
-
-
Abstract:
1) The abstract must provide information about the research method (a more detailed explanation is needed regarding the research design, sample selection, and data collection tools.
2) The abstract should provide a clearer picture of the research results, especially the implications of conceptual changes experienced by veteran teachers. This will give readers a better understanding of the impact of using different teaching methods.
3) Arrange keywords alphabetically.
Introduction:
1) Clear research gaps are essential for readers to understand the importance of this research.
2) It is important to provide a comprehensive overview of existing research to establish context and justify current research needs.
3) A clear statement of the research question is required. Well-formulated research questions help guide the research and give readers an idea of the research objectives.
Literature review: okay
Method:
1) The participants in this research were only three of the eight teachers who took part in in-service training. Small sample sizes can limit the ability of research findings to be replicated. Including information about how participants were selected can provide further insight into their representativeness.
2) The explanation of the image analysis process and accompanying text is quite short. More details about the criteria used in the analysis and the decision-making process in interpreting the results could increase transparency and strengthen the validity of the analysis.
3) Although “multi-modal text” is used, there is no in-depth explanation of how this approach is implemented and how aspects such as the relative size of images or direction of interaction are evaluated. Further explanation regarding the application of this method will increase the reader's understanding.
4) The use of references in explaining methods can be improved. Including additional references that support the selection of analytical methods and tools can provide a stronger theoretical foundation for this research.
Result: okay
Discussion:
1) Provide an in-depth interpretation of your research results.
2) Can you compare with previous research, are your results consistent or different from previous research?
3) Can you explain the implications of your research findings in a practical and theoretical context?
Conclusion: no conclusion
1) The conclusion section of a research article plays an important role in summarizing the main findings, evaluating their significance, and presenting practical implications and suggestions for future research.
2) Summarize the main findings of your research. Can you identify the results that are most significant and relevant to the research objectives?
3) Relate your findings to the research objectives and research questions.
4) Present Practical Implications: Explain how research findings can be applied in a practical context. Identify potential implications for policy, practice, or society.
5) Provide suggestions and recommendations for further research. Can you identify areas that still require further exploration or understanding?
-
Author Response
Please see the attached responses.

Reviewer 4 Report
Comments and Suggestions for Authors
The detailed breakdown of different aspects, such as research design, questions, hypotheses, methods, arguments, coherence, empirical results, and engagement with sources, is a valuable structure for evaluating the paper. The positive aspects include the detailed descriptions of the participants' drawings, the evolution of their perspectives, and the impact of the Attentive Teaching method on their conceptualizations
Strengthen the engagement with existing literature by providing more explicit connections between the study's findings and previous research. Highlight how your study contributes to or challenges existing knowledge.
Ensure that the conclusions are well-supported by the results and any referenced secondary literature. Summarize how the study addresses the research questions and contributes to the field..
It is necessary to clearly articulate the originality of your study and how it contributes to the existing body of knowledge on teaching concepts. What unique insights does your research offer?
Comments on the Quality of English Language
Clarity:
The language appears to be clear, and sentences are generally well-structured, making it easy for the reader to follow the narrative.
Coherence:The language appears to be clear, and sentences are generally well-structured, making it easy for the reader to follow the narrative.
The writing exhibits coherence, with ideas flowing logically from one point to the next. Transitions between sections and paragraphs seem effective.
Academic Tone:
The tone is academic and appropriate for a research paper. The language used is formal and reflects a scholarly approach.
Grammar and Syntax:
Based on the provided sections, there don't seem to be significant issues with grammar or syntax. However, a thorough proofreading might catch any minor errors that could be present.
Technical Terminology:
The use of technical terminology related to the subject matter appears accurate and contextually appropriate.
Conciseness:
The text seems concise and avoids unnecessary verbosity. This is generally positive for academic writing.
It is also recommended to have a colleague or someone proficient in academic writing review the paper for additional perspectives. Overall, it seems that the English language quality is appropriate for an academic research paper.
Round 2
Reviewer 3 Report
Comments and Suggestions for Authors
Thank you for taking the time and effort to correct and perfect the manuscript according to the feedback given by the reviewers. Your insights and improvements to the manuscript have contributed to improving your work and can enhance the strength and credibility of our journal. I appreciate the thorough improvements to this article.
I have carefully read each point proposed in the related corrections to improve the manuscript's clarity, coherence, and overall quality.
Once again, we thank you for your efforts to improve this manuscript. The revised manuscript now better serves its purpose and will make a meaningful contribution to the field.
Thank You
Regards,